# communications
## earth & environment

# Global patterns and predictors of C:N:P in marine ecosystems

Tatsuro Tanioka [1], Catherine A. Garcia[1,2], Alyse A. Larkin [1], Nathan S. Garcia[1], Adam J. Fagan[1] & Adam C. Martiny [1,3✉]

Oceanic nutrient cycles are coupled, yet carbon-nitrogen-phosphorus (C:N:P) stoichiometry in marine ecosystems is variable through space and time, with no clear consensus on the controls on variability. Here, we analyze hydrographic, plankton genomic diversity, and particulate organic matter data from 1970 stations sampled during a global ocean observation program (Bio-GO-SHIP) to investigate the biogeography of surface ocean particulate organic matter stoichiometry. We find latitudinal variability in C:N:P stoichiometry, with surface temperature and macronutrient availability as strong predictors of stoichiometry at high latitudes. Genomic observations indicated community nutrient stress and suggested that nutrient supply rate and nitrogen-versus-phosphorus stress are predictive of hemispheric and regional variations in stoichiometry. Our data-derived statistical model suggests that C:P and N:P ratios will increase at high latitudes in the future, however, changes at low latitudes are uncertain. Our findings suggest systematic regulation of elemental stoichiometry among ocean ecosystems, but that future changes remain highly uncertain.

[1] Department of Earth System Science, University of California Irvine, Irvine, CA, USA. [2] Center for Microbial Oceanography: Research and Education (C-MORE), University of Hawaii at Manoa, Honolulu, HI, USA. [3] Department of Ecology and Evolutionary Biology, University of California Irvine, Irvine, CA, USA. ✉email: amartiny@uci.edu

Carbon-Nitrogen-Phosphorus (CNP) stoichiometry is widely used in oceanographic studies to provide critical linkages between the availability of key nutrients, primary productivity, and carbon sequestration[1,2]. C:P, N:P, and C:N ratios of suspended particulate organic matter (POM) in the surface ocean, reflecting the ecosystem elemental composition, vary systematically between regions. The ratios are commonly below the canonical Redfield ratio of 106, 16, and 6.7, respectively, in the cold, nutrient replete high-latitude regions and above the Redfield ratios in the warm, nutrient deplete subtropical gyres[3,4]. Observed C:N:P ratios also display temporal variability on daily[5,6], seasonal[7], and inter-annual timescales[8,9]. As changes in C:N:P ratios can have cascading effects on the carbon cycle[10,11], nitrogen cycle[12,13], and marine food-web dynamics[14], identifying the environmental drivers of C:N:P has become a pressing challenge.

There are several alternate, although not necessarily mutually exclusive hypotheses for mechanisms controlling the C:N:P of suspended POM in marine ecosystems[15–17]. Temperature and nutrients can modulate cellular C:N:P of phytoplankton on the timescales of days to weeks[18,19]. Furthermore, change in the plankton biodiversity from selection to temperature and nutrient variations can alter bulk ecosystem C:N:P[20,21] because different taxonomic lineages of plankton may have unique optimal C:N:P[22]. The challenge is that the relative importance of temperature versus nutrients is not currently well quantified, stemming from limited spatial coverage and the dearth of direct measurements for nutrient stress experienced by plankton communities in mid-low latitude oligotrophic regions[11,23,24]. Previous global synthesis studies[3,11] relied on dissolved nitrate and phosphorus concentrations to measure nutrient stress, but nutrients are often below analytical detection limits in many low latitude ecosystems[24], prohibiting accurate diagnosis of N vs. P limitation[25]. The nutrient limitation type (e.g., N vs. P limitation) is critical as phytoplankton C:P and N:P cellular ratios can vary by as much as a factor of three between P-limited and N-limited conditions under otherwise the same growth environment[26,27]. As a result of these shortcomings, we still lack a quantitative understanding of what drives marine ecosystem C:N:P stoichiometry.

Here, we quantify the global variation and identify key environmental predictors for surface ocean ecosystem C:N:P. We collected and analyzed POM samples across all major ocean basins as part of the biological initiative for the Global Ocean Ship-based Hydrographic Investigations Program or Bio-GO-SHIP[28,29]. The Bio-GO-SHIP dataset greatly expanded the spatial coverage from previous global CNP studies[3,11,30] (Fig. 1) and now includes samples from regions like the South Subtropical Pacific, South Atlantic, and the Indian Ocean. We identified relationships between C:N:P and diverse environmental predictors, including phytoplankton nutrient stress, from paired metagenomics observations[31] (Supplementary Fig. 1). Finally, we applied our data-derived statistical models to the output from the Community Earth System Model Large Ensemble Simulation (CESM2-LENS)[32] to project surface ecosystem C:N:P for the historical period (years, 2010–2014) and end of the 21st century (years, 2095–2100, shared socioeconomic pathways SSP3-7.0) to identify areas that may undergo the most drastic change in ocean elemental stoichiometry. SSP3-7.0 scenario is the second most pessimistic, high-greenhouse-gas emission trajectory[33], where $CO_2$ doubles compared to pre-industrial by 2100 and radiative forcing level reaches 7.0 W/m². Our projections from the data-derived statistical model show consistent increases in C:P and N:P under the future climate scenario in the high latitude ecosystems, which agrees with projections made by Earth system models[14,34,35]. However, projections made by two modeling approaches diverge considerably in lower latitude ecosystems, indicating that future changes in C:N:P, especially at low latitudes remain highly uncertain.

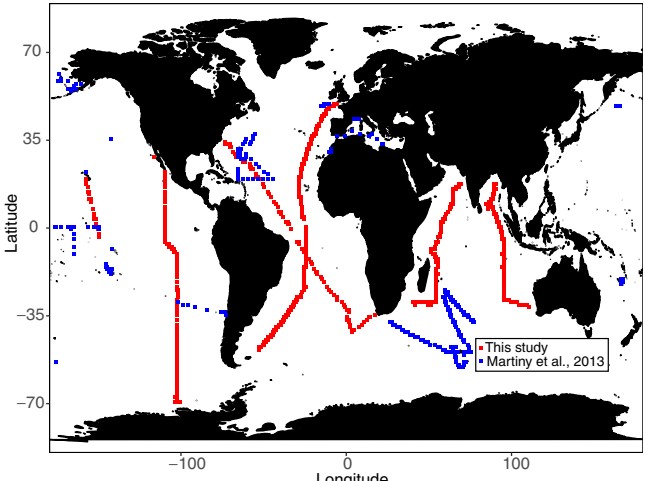

**Fig. 1 Geographical sampling stations of particulate organic matter in the global ocean.** Red points are stations from Bio-GO-SHIP ($n = 1970$) and blue points are from a previous global compilation[3] ($n = 733$).

The data-driven statistical approach, which first establishes relationships among C:N:P and environmental factors along contemporary ocean environmental gradients and then applies the same statistical relationship to the future environmental condition, is an alternative to Earth system models for predicting future changes to C:N:P. Although data-driven statistical approaches lack a mechanistic basis, they can integrate poorly understood biological mechanisms. For example, this approach implicitly embraces the plankton diversity, interactions between different environmental factors, and poorly understood biotic effects of higher trophic levels[36]. Earth system models, on the other hand, are mechanistic and anchored in theory but often rely on simplistic assumptions and parametrizations owing to our incomplete understanding of biological systems. Divergent future projections amongst the two modeling approaches in low latitude ecosystems suggest that there are critical knowledge gaps for the regulation of C:N:P.

## Results

We collected 1970 paired POM samples (C, N, and P) in the top 30 m across a broad latitudinal range from 70 °S to 50 °N (Fig. 1, Supplementary Table 1) and analyzed them using consistent protocols. The global area-weighted mean C:N:P was 137:21:1 (Supplementary Table 2, 3), which largely agrees with a previous data compilation of surface ecosystem C:N:P of 146:20:1[3]. Ecosystem C:N:P ratios exhibited a robust latitudinal pattern, highest in the subtropical gyres, intermediate in equatorial regions, and low towards higher latitudes (Fig. 2, Supplementary Table 4). The highest C:P and N:P were observed in the western North Atlantic, where mean values reached 225 and 32, respectively. The lowest values were observed in areas poleward of the Southern subtropical convergence, with the lowest observed C:P and N:P ratios of ~60 and ~10, respectively. The latitudinal trends in C:P and N:P were mirrored in both hemispheres, but peak C:P and N:P ratios were commonly higher in the Northern vs. Southern Hemisphere. C:N was close to the canonical Redfield ratio of 6.6 in most regions but noticeably elevated in the eastern parts of the southern subtropical gyres in the Atlantic, Indian, and Pacific Oceans, with C:N exceeding 8. In contrast, C:N was slightly lower than the Redfield ratio in the Southern Ocean, with a mean of ~6. Thus, C:N:P showed a latitudinal gradient and clear hemispheric and longitudinal deviations.

To identify environmental predictors of C:N:P, we conducted a combination of correlation analysis and analyses using generalized

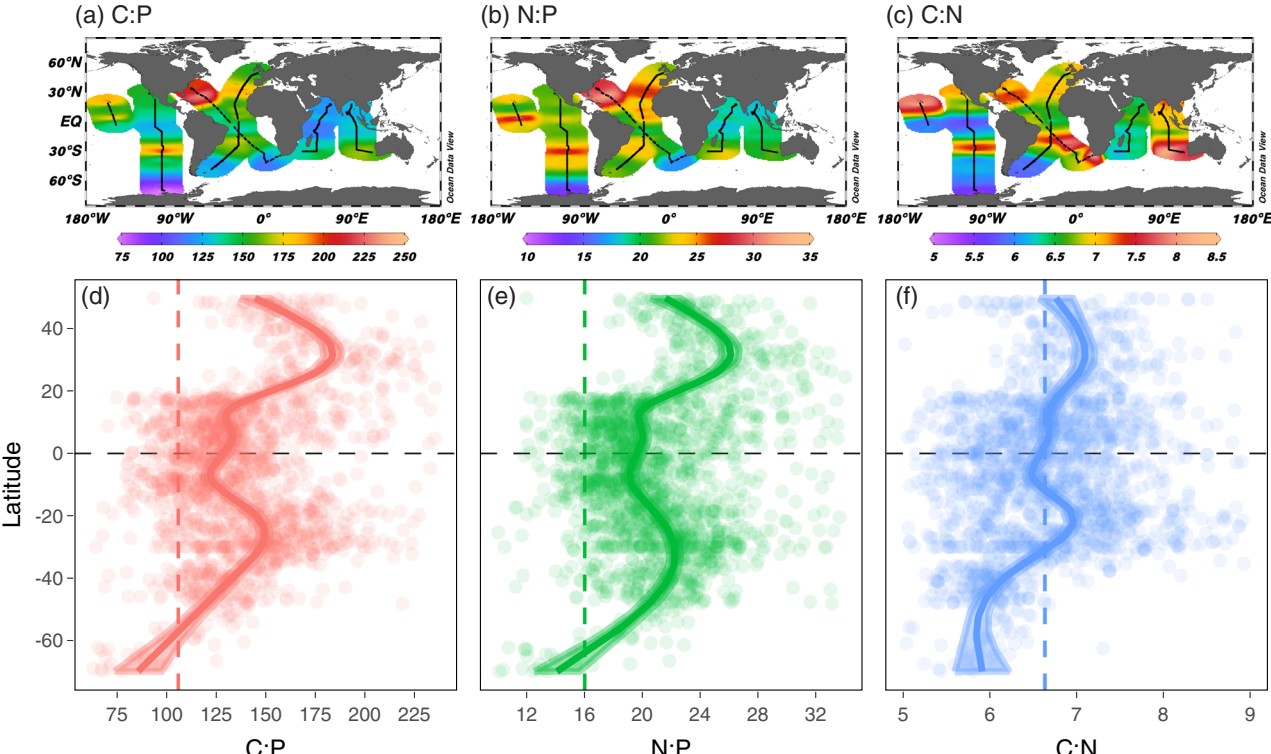

**Fig. 2 Global distribution and latitudinal trends of surface ecosystem C:N:P. a–c** Individual sampling locations are shown with black points in the global map of C:P, N:P, and C:N. Multi-color shadings in **a–c** are based on weighted-average gridding from Ocean Data View. **d–f** Measurements of C:P, N:P, and C:N are plotted against latitude and solid lines represent the Generalized Additive Model (GAM) smooth trends and ribbons corresponding to the 95% confidence intervals of latitudinal trends predicted by the GAMs. The dotted vertical lines show the canonical C:N:P Redfield ratio of 106:16:1.

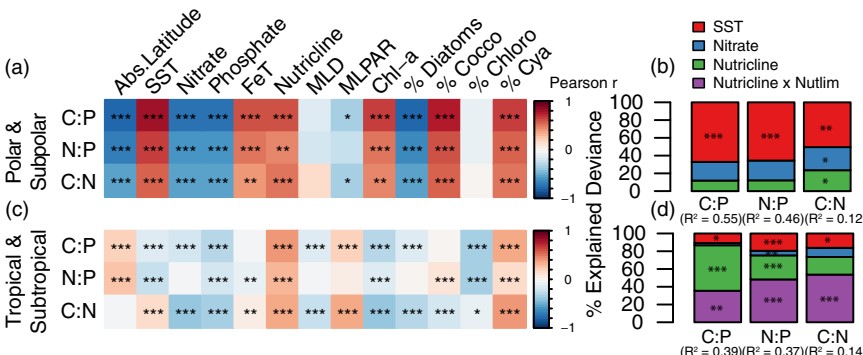

**Fig. 3 Predictors of ecosystem C:N:P. a, c** Correlation of contextual variables with the C:N:P ratios. The color of the tiles is the Pearson $r$ correlation coefficient. Asterisks represent the statistical significance (***$p < 0.001$, **$p < 0.01$, *: $p < 0.05$). **b, d** The individual explained deviance and additive contribution of the four main contextual variables normalized to the total explained deviance in GAMs. The bracket number is deviance explained ($R^2$), by the full model, which equals the sum of deviance explained by the individual variable. **a, b** corresponds to the data collected in the (sub)polar regions with |Latitude| ≥ 45° ($n = 145$), and the **c**, and **d** corresponds to the data collected in the (sub)tropical regions with |Latitude| < 45° ($n = 1825$).

additive models (GAMs). While the correlation analysis can capture first-order, monotonic relationships between predictors and C:N:P, GAMs detected nonlinear, non-monotonic relationships amongst C:N:P and in situ measurements of sea surface temperature (SST), nutrient availability, and nutrient limitation type. Nutricline depth (here defined as the depth at which nitrate concentration equals 1 μmol kg$^{-1}$) is used as a proxy of nutrient supply rate, where deeper nutricline indicates a lower nutrient supply rate to the upper mixed layer of the ocean[37]. Overall, we found that the dominant environmental predictors of surface ecosystem C:N:P differed between high and low-latitude regions (Fig. 3). In (sub)polar regions, SST was strongly positively correlated with C:P and N:P (Fig. 3a, Supplementary Table 5, 6), and SST captured 67% and

65% of the total explained variances for C:P ($R^2 = 0.55$), and N:P ($R^2 = 0.46$), respectively (Fig. 3b, Supplementary Table 7). C:P and N:P increased linearly from the coldest polar regions to the warmer subpolar regions, coinciding with a gradual community composition shift from diatom to coccolithophore dominance (Fig. 3a). Here, phytoplankton-group relative abundance was obtained from the NASA Ocean Biogeochemical Model[38,39] at the closest grid point to the spatial position of each POM sampling point. Nitrate and phosphate concentrations were significantly negatively correlated with C:N:P across high latitudes, but macronutrient concentrations were not as good of a predictor for C:N:P as SST (Fig. 3b, Supplementary Fig. 2). Nutricline could not explain variances in C:N:P as the surface nitrate concentrations exceeded

$1 \, \mu\text{mol kg}^{-1}$ in large parts of the high latitude ecosystems. Similarly, the element-specific nutrient stress (i.e., N vs. P vs. Fe stress) could not explain C:N:P variability in the high latitudes because regions from which samples were collected were uniformly Fe-limited (Supplementary Fig. 1a). To summarize, temperature and macronutrient availability were primary predictors of C:N:P variability in high latitudes, coinciding with a noticeable shift in the phytoplankton community through fractional decreases in diatom and the concomitant increases in coccolithophore and cyanobacteria abundances.

In (sub)tropical ecosystems, nutricline depth and the element-specific nutrient stress were the strongest environmental predictors for C:N:P. In these warm regions, we observed that 77–87% of the explained variance for C:N:P was attributed to the nutricline depth plus element-specific nutrient stress (Fig. 3d). However, total deviance explained by GAM was noticeably lower in the low latitude ecosystems ($R^2 = 0.39$, 0.37, and 0.14 for C:P, N:P, and C:N) than in the high latitude ecosystems (Supplementary Table 8). Without considering nutrient stress, GAMs predicted that C:P and N:P increased monotonically with warming until ~20 °C and then plateaued (Fig. 4, Supplementary Fig. 3a). C:P and N:P were highest with interaction with a deep nutricline and P-stress or P/N co-stress (Fig. 4b, Supplementary Figure 3b). C:N was highest when the nutricline was deep and phytoplankton were N-stressed (Fig. 4d). Regardless of nutrient limitation types, C:P, N:P, and C:N converged to similar values of 125, 18, and 6.7, respectively, when nutricline depth approached 0 m and thus where nitrate remained abundant at the surface. Nitrate and phosphate concentrations explained little C:N:P variability as macronutrient concentrations were at or below detection limits across most low latitude sites (Supplementary Fig. 2). In summary, the global synthesis of surface ecosystem C:N:P revealed a transition from a temperature and macronutrient dependency at high latitudes to a multidimensional nutrient stress control in mid-to-low latitudes.

We next projected the present and future global distributions of surface C:P and N:P stoichiometry. These projections were made by combining the observation-constrained GAMs with projections of present and future oceanic conditions under shared socioeconomic pathways SSP3-7.0 scenario (Fig. 5, Supplementary Fig. 4). We predicted a general future increase in C:P at high latitudes but a decrease in the subtropics and tropics (Fig. 5c). This spatial pattern was similar for N:P (Supplementary Fig. 5). Overall, the global area-weighted mean C:N:P changed little from 120:19:1 in the 2010s to 124:19:1 in the 2090s (Supplementary Table 9). However, the area-weighted mean C:P poleward of 45° increased from 83 in the 2010s to 94 in the 2090s. This high latitude increase was predominantly due to a 2–3 °C warming (Supplementary Fig. 5) and largely agrees with projections made by fully prognostic ocean biogeochemical models (Supplementary Figure 6). In the mid-low latitudes (equatorward of 45°), our data-driven statistical model projected an overall constant C:P. However, there are large geographical differences leading to regions with strong declines (e.g., western North Atlantic due to a shoaling nutricline) or increases (e.g., western North Pacific shifting to P-limitation and South Pacific with a deepening nutricline). Moreover, model agreement, which reflects the predictability of C:P by the data-derived statistical model, rarely exceeded 70% in the mid-low latitudes (Fig. 5b, d). Regions with the lowest model C:P predictability corresponded to areas with the smallest projected change in C:P, such as the boundary between subpolar and subtropics, where the annual mean SST was 15–20 °C. Similarly, projections from biogeochemical models are not in agreement with each other in low latitude ecosystems (Supplementary Fig. 6). To summarize, independent model projections made by data-derived approach and mechanistic

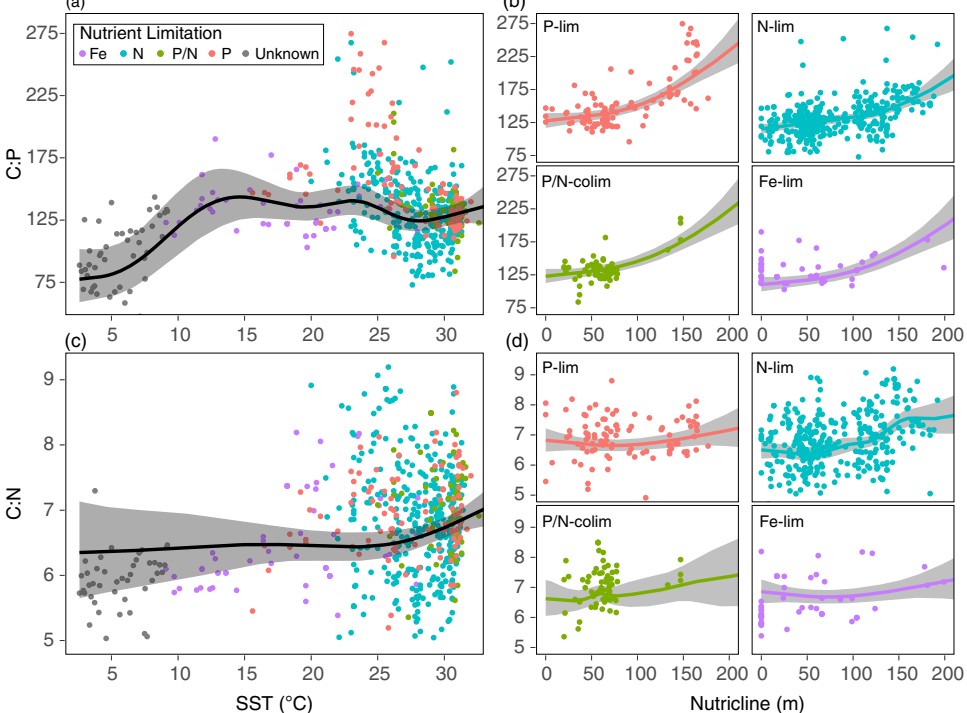

**Fig. 4 Observed C:P and C:N as a function of environmental variation.** Dots are observed values and colors represent the nutrient limitation type inferred from metagenomes (Purple = Fe-limited, Blue = N-limited, Green = P/N co-limited, Red = P-limited, Grey = Unknown). **a**, **c** C:P and C:N against SST. Black line and shade represent GAM prediction and uncertainty (± 2SE) under the constant nutricline depth and surface nitrate values at the observed mean values of 70 m and $0.2 \, \mu\text{mol kg}^{-1}$, respectively. **b**, **d** C:P and C:N against nutricline depth for different nutrient limitation types. GAM is fitted separately for each limiting nutrient type under constant SST and surface nitrate at the observed mean values of 25 °C and $0.2 \, \mu\text{mol kg}^{-1}$, respectively.

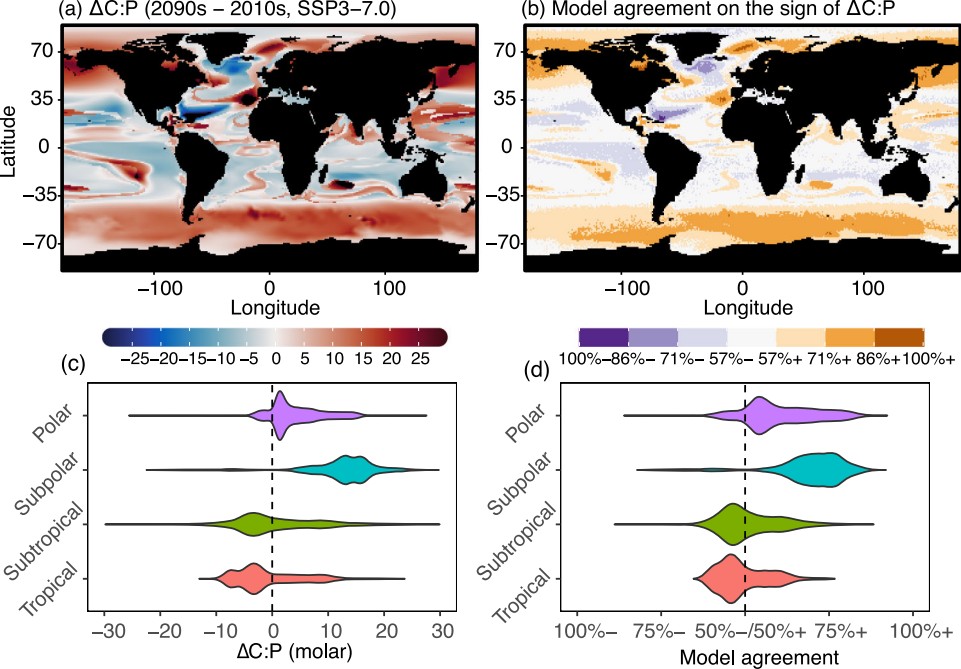

**Fig. 5 Projected surface ecosystem C:P using a data-derived statistical model. a** Difference in surface ecosystem C:P estimated for the 2090 s and 2010s projected using a data-derived statistical model coupled to sea surface temperature, surface nitrate concentration, nutricline, and nutrient limitation type of small phytoplankton from CESM2-LENS under the shared socioeconomic pathways SSP3-7.0 and historic scenarios, respectively. **b** Model agreement on the sign of change in C:P amongst 2000 randomly generated model projections based on the posterior distribution of the GAM parameters. 100%+ represents the case when all 2000 models predict the positive change in C:P, and 100% represents the case when all models predict the negative change in C:P. Note that 50%+/50%- corresponds to the minimum agreement between 2000 models. Violin plots for change in **c** C:P and **d** model agreement for regions separated by latitude. Regions: Polar (|Latitude| ≥ 65°), Subpolar (45° ≤ |Latitude| < 65°), Subtropical (15° ≤ |Latitude| < 45°), and Tropical (|Latitude| < 15°).

approaches suggest an increase in C:P and N:P in the high latitude ecosystems but changes in low latitude ecosystems remain uncertain under the future climate scenario.

## Discussion

Our global analysis supports a link between temperature, surface nutrient depletion, and N vs. P stress with C:N:P stoichiometry. A strong temperature dependency of C:P and N:P in high latitude ecosystems is consistent with the translation compensation hypothesis[17,40], where plankton increase allocation to P-rich ribosomes for biosynthesis at low temperature, leading to lower C:P and N:P. Lower temperature also leads to lower C:N of phytoplankton by slowing down the metabolism of phytoplankton and decreasing their ability to consume nitrate, thus increasing residual nitrate concentrations[41]. The transition from a strong temperature dependency at higher latitudes to a strong nutrient dependency at low latitudes may be due to a weakened temperature control on phytoplankton growth under low nutrient supply rate conditions[42,43]. Thus, our data support the translation compensation hypothesis and the strong temperature dependency on C:N:P but only in nutrient-replete environments. This study did not consider the effect of temperature in cold regions that are depleted in surface macronutrients. Therefore we suggest expanding sample coverage to the Arctic Ocean to understand further how low temperature affects C:N:P.

In low latitude ecosystems, our global data suggest C:N:P is regulated to a large extent by an interaction between the overall nutrient supply and the elemental nutrient stress type. There is compelling support in theoretical and lab culture experiments for this multi-dimensional nutrient control of C:N:P. Chemostat models predict a more flexible stoichiometry of phytoplankton cells at lower nutrient supply and growth but a fixed C:N:P at

$\mu_{max}$[26,44]. Similarly, culture experiments show that cellular C:N:P is very sensitive to N vs. P stress at low growth rates, but this flexibility narrows with higher growth rates[27,45]. Although we cannot directly measure nutrient supply, a deeper nutricline likely reflects a lower overall nutrient supply rate[37]. Thus, the observed interactive relationships between C:N:P, nutricline depth, and N vs. P stress seem to align well with these theoretical and laboratory culture predictions.

An inter-hemisphere contrast in ecosystem C:N:P in low latitude ecosystems may be linked to differences in the N:P:Fe supply ratio and the relative degree of N vs. P stress[5]. More pronounced C:P and N:P peaks are observed in the Northern vs. Southern Hemisphere subtropical gyres. We associate the higher ecosystem C:P and N:P in the Northern Hemisphere with a more substantial surface phosphate depletion in the North Atlantic and Pacific gyres from the higher Fe supply and $N_2$ fixation[24]. In contrast, we more commonly observed regions of high C:N in the Southern Hemisphere, including the eastern South Atlantic, eastern South Pacific, and eastern South Indian Oceans. These are strongly N-stressed regions with depressed Fe supply and $N_2$ fixation[12,46,47]. In addition to cellular level changes in C:N:P, low latitude ecosystems typically favor slow-growing cyanobacteria with higher C:P and N:P ratios over eukaryotes with lower stoichiometric ratios[20,48]. Indeed, we globally observed a significant positive correlation between C:P and N:P with % cyanobacteria and a negative correlation with % diatoms (Fig. 3a, c). However, hemisphere differences in C:N:P rule out that community shifts alone control the observed C:N:P. In summary, nutrient supply rate and ratios are potentially the best predictors of large C:N:P variability in low latitude marine ecosystems, while temperature and macronutrient availability seem to shape the overall latitudinal gradient.

We observe a mild decrease in C:P and N:P in low latitude ecosystems at high temperatures above 20 °C. This decrease in C:P and N:P may be related to an increase in cellular RNA content to meet a greater demand of chaperones required for the repair of heat-induced damage[18] or to the disproportionate increase in the respiration over photosynthesis leading to lower carbon fixation at higher temperature[49]. However, we currently lack the observations from regions with a surface temperature above 30 °C to fully constrain the relationship between warming and C:N:P leading to uncertain model projections. Thus, we suggest sampling in extremely warm regions like the western Pacific Ocean or marginal seas with a surface temperature above 30 °C, providing analog conditions for a future warm world.

There are several important caveats to our observation and the data-driven statistical approach for projecting C:N:P. First, data-driven statistical models assume that plankton physiology and community will share the same relationship to environmental conditions in the present and future ocean. These projections incur considerable uncertainties when extrapolating the statistical models outside the currently observed/observable state of the system. Second, we did not consider the roles of dissolved organic matter. Plankton's ability to access dissolved organic matter, particularly at high temperatures, may be an important driver for shifting the balance between C, N, and P in areas such as North Atlantic and western South and North Pacific[50]. However, dissolved organic matter is chemically diverse[51], and we were unable to incorporate it as a predictor here. Thirdly, we solely used *Prochlorococcus* genomes to diagnose nutrient stress for the plankton community. As *Prochlorococcus* make up a large percentage of community biomass in the tropics and subtropics[52], their physiological status is likely important for the total phytoplankton community. However, in regions with lower *Prochlorococcus* abundance, other lineages are likely important for the ecosystem state and may deviate from *Prochlorococcus*. Fourth, a change in the nutrient supply ratio could lead to an abrupt shift in plankton community composition[53], which in turn may abruptly shift the ecosystem C:N:P. Such changes in nutrient supply ratios may be driven by anthropogenic N emission[54], shifting nitrogen fixation[55], and atmospheric nutrient deposition[56]. As these abrupt ecological shifts are expected to precede early warning signals from temperature and nutrients[53], it is critical to expand monitoring of ecosystem C:N:P through long-term monitoring[7,57], shipboard measurements[29], and remote sensing[58]. These spatial and temporal sampling efforts are critical for narrowing down the degree of uncertainty in model projections of C:N:P.

## Methods

**POM sample collection**. In this study, we use paired observations of particulate organic phosphorus (POP), nitrogen (PON), and carbon (POC) samples from 1970 stations collected between 2014 and 2020 as a part of a biological initiative for the Global Ocean Ship-Based Hydrographic Investigations Program (Bio-GO-SHIP)[28,29]. Samples used in this study are from cruises AMT-28, C13.5, I07N, I09N, NH1418, and P18 (Supplementary Table 1). Samples were collected across all major oceanic provinces from 70 °S to 50 °N using the consistent sampling method described previously[5,28,59,60]. Briefly, 2–10 L seawater for the POM samples was collected from the onboard flow-through underway system at the sea surface (<30 m) and was divided into POC/PON and POP triplicates after removing large plankton and particles using 30 μm nylon mesh. Each replicate was then filtered on precombusted Whatman GF/F filters with a nominal pore size of 0.7 μm. POP filters were rinsed with 5 mL of 0.17 M Na₂SO₄ prior to analysis to remove traces of dissolved organic phosphorus. All filtered POM samples were sealed in pre-combusted aluminum packets and were immediately frozen at −20 °C until analysis.

POC and PON samples were measured using Control Equipment 240-XA/440-XA elemental analyzer standardized to acetanilide or a CN Flash 1112 EA elemental analyzer against an atropine (C₁₇H₂₃NO₃) standard curve. The POC analysis included an acidification step in concentrated HCl fumes to remove particulate inorganic carbonates. POC and PON measurements had a mean detection limit of ~2.4 μg and ~3.0 μg, respectively. POP was analyzed

using the ash-hydrolysis colorimetric method described previously[61] using a spectrophotometer at 885 nm. The detection limit for POP measurement was ~0.3 μg.

Following the criteria used in a previous study[60], we discarded any anomalous samples with POC:POP > 500, PON:POP < 1, and PON:POP > 100 after the stoichiometric ratios were calculated. These selection processes led to the 1970 final C-N-P paired POM measurements. To evaluate the influence of spatial autocorrelation, we binned the samples into 1° by 1° grid cell and computed globally area-weighted values with this dataset. Our analysis showed that the global area-weighted means of binned and unbinned data are indistinguishable and concluded that such spatial autocorrelation was not a problem in our data analysis (Supplementary Table 2, 3). Based on previous studies[3,30], a large proportion of POM pools collected are assumed to be made up of living planktonic materials consisting of *Prochlorococcus*, *Synechococcus*, eukaryotic phytoplankton, and bacteria with a minor contribution from microzooplankton and heterotrophic nanoflagellates.

**Hydrography measurements**. Hydrographic measurements (salinity, temperature, and pressure) were taken at each station with a CTD-rosette vertical profiling system. Ambient nitrate, phosphate, and silicate concentrations were determined onboard using an auto-analyzer following the GO-SHIP nutrient protocol[62] for cruises AMT-28, C13.5, I07N, I09N, and P18. Macronutrients (N or P) in cruise NH1418 were measured in the lab[63], and the detection limits were 0.05 μmol kg⁻¹. Bottle data for macronutrients were linearly extrapolated horizontally where necessary to match the sampling resolution of underway data (i.e., POM data). For the C13.5 section in which in situ nutrient measurements were not measured due to COVID-19 related logistical issues, we substituted missing values with mapped annual mean average values from the GLODAP version2.2016b from the nearest longitude and latitude at 1° resolution[64,65]. We set consistent detection limits for phosphate and nitrate at 0.01 and 0.1 μmol kg⁻¹, respectively, for all the hydrographic measurements and corrected any measured concentrations below these values are assumed to be equal to the threshold concentrations for use in statistical analysis. Nutricline depth, here defined as the depth at which nitrate equals 1 μmol kg⁻¹, was determined by vertically and horizontally interpolating nitrate concentration. We set nutricline as 0 m when the bottle nitrate concentration at the shallowest depth was greater than 1 μmol kg⁻¹. Previous studies[37,66] have revealed that nutricline depth, where deeper nutricline indicates a lower nutrient supply rate to the upper mixed layer of the ocean, serves as a good proxy for an overall nutrient supply rate in the surface water than ambient macronutrient concentrations, which are often at detection limits.

**Contextual environmental variables**. We complemented in situ measurements with (i) mixed-layer averaged photosynthetically available radiation (PAR)[67], which was estimated using surface PAR, Chl-a, and monthly climatology of mixed layer depth[68], (ii) the average phytoplankton community composition (diatoms, coccolithophores, chlorophyte, and cyanobacteria) between 1998–2017, which we obtained from NASA Ocean Biogeochemical Model[38,39], and (iii) the annual mean total dissolved iron, which we derived from Community Earth System Model v1.2.1. Both NASA Ocean Biogeochemical Model and CESM were calibrated with observations and have been used extensively in previous global biogeochemistry studies[20,31]. The model phytoplankton community composition from NASA Ocean Biogeochemical Model only exists from 1998 to 2017. For data from 2018 onwards, we used the model output from 2004, which is the year with the minimum sum of deviations from the monthly mean, following the previous study[20]. PAR and Chl-a are 8-day averaged values retrieved by NASA MODIS-Aqua at the nearest location (4 km resolution) (http://oceancolor.gsfc.nasa.gov (last access: July 29, 2021)). Climatological mixed layer depth is derived from more than 1.2 million Argo profiles[68] and provides accurate information about the seasonal patterns of global mixed layer depth.

**Metagenomics-informed nutrient limitation**. We used the previously published global genome content of *Prochlorococcus* and its inferred element-specific nutrient stress[31]. Specifically, we selected data from 562 stations, where metagenome samples were collected concomitantly with POM (Supplementary Fig. 1). We used metagenome samples collected in the regions encompassing 51.5 °S and 47.9 °N, where the abundance of *Prochlorococcus* was sufficient. Briefly, sequences from the surface metagenomes were recruited to known strains of *Prochlorococcus*, and the frequency of established nutrient acquisition genes determined a priori were used as a proxy for nutrient stress type (i.e., limiting nutrient element) and severity. For example, the presence of marker genes *phoX* and *phoA*, responsible for regulating alkaline phosphatases required for the assimilation of dissolved organic P, are associated with high phosphorus stress. A previous study has shown a significant correlation between *Prochlorococcus* nutrient stress index and growth/turnover rate from nutrient bottle incubation experiments[31]. An ordination of nutrient genes based on the angles from the principal component analysis can broadly categorize six types of limitation and co-limitation: (1) Fe limitation, (2) Fe/P co-limitation, (3) P limitation, (4) P/N co-limitation, (5) N limitation, and (6) N/Fe co-limitation. As the number of samples for Fe/P co-limitation and N/Fe co-limitation samples was noticeably smaller than other stress types, we merged Fe/P and N/Fe with P

and N limitation samples, respectively. Our dataset consists of 101 P-limitation samples, 337 N-limitation samples, 67 P/N co-limitation samples, and 57 Fe-limitation samples that are geographically and temporally paired with POM samples. The global map of nutrient limitation from metagenomes is largely consistent with the nutrient limitation pattern of the small phytoplankton from the CESM model output (Supplementary Fig. 1).

**Data analysis and modeling**. All the statistical analyses were conducted using R ver. 4.1.0[69]. To determine the relative importance of different contextual variables required to explain C:N:P, we first conducted multiple pairwise correlation analyses using the Pearson correlation test, which allowed us to determine a first-order linear relationship between a covariate and C:N:P. We used natural log-transformed values of elemental stoichiometric ratios and nutrient concentrations throughout the data analysis. For fair comparison across variables, we removed any rows containing the missing value from the dataset and standardized all the variables so that the mean equaled zero and the standard deviation equaled one. We correlated C:N:P with various environmental drivers including in situ measurements of SST, surface phosphate, surface nitrate, and nutricline depth; mixed-layer depth, mixed-layer averaged PAR, nutricline depth, modeled surface plankton community composition (% Diatoms, % Coccolithophores, % Chlorophytes, % Cyanobacteria), and total dissolved iron from the model simulations (Supplementary Table 5, 6). We performed separate analyses for the (1) polar/subpolar ($n = 145$) and (2) tropical/subtropical regions ($n = 1825$) which were delineated based on the absolute latitude of 45°.

We subsequently conducted analyses with generalized additive models (GAMs) to identify the relative strength of four main environmental variables in explaining C:N:P ratios: these were (1) SST, (2) surface nitrate concentration, (3) nutricline depth, and (4) the limiting nutrient type of *Prochlorococcus* determined from the metagenome analysis. We chose these variables based on the correlation analysis and the previous understanding of ecological stoichiometry. For the GAM analysis, we used the R package *mgcv*[70]. For GAM analyses in (sub)tropical regions, we used the subset of POM data where both POM and metagenomes were collected ($n = 554$). We conducted cross-validation (100 random partitions holding out 20% of observations) on different possible hierarchical GAM formulations[71]: (1) Model G (A global smoother for all observations), (2) Model GS (Single common smoother plus group-level smoothers that have the same wiggliness), (3) Model GI (Single common smoother plus group-level smoothers that have the different wiggliness), (4) Model S (Group-specific smoothers without a global smoother, but all smoothers have the same wiggliness), (5) Model I (Group-specific smoothers with different wiggliness), and (6) Model C (Control, no dependence on nutrient limitation types) (Supplementary Methods). We found that the models with the interactive effect of nutricline and element-specific nutrient limitation type (model GI and I) outperformed the models with either independent (model G) or null effects (model C) of nutrient limitation type in terms of Akaike information criterion, root-mean-square error, and the coefficient of determination (Supplementary Tables 10–12). Specifically, the model GI performed best out of all the possible model types of functional variation for hierarchical GAM. Thus, we decided to use the model GI to describe the interaction between nutricline and element-specific nutrient limitations throughout the paper. The additive contribution of each contextual variable (SST, nitrate, nutricline, and the interaction between nutricline and nutrient limitation type) to the total deviance explained was calculated by sequentially removing different parameters and averaging sequential sums of squares over all ordering of regressors before normalizing with deviance explained by a null model. This approach ensures that the sum of each regressor's deviance explained adds up to the full model deviance explained[72].

We repeated GAM analyses with the previous global C:N:P compilation[3] binned by longitude and latitude at 1° resolution ($n = 204$), combined with SST, nitrate, and nutricline depth from GLODAP version2.2016b[64,65] and small phytoplankton nutrient limitation pattern from CESM2 Large Ensemble Simulation at the 2010s. We found the overall consistency in the explained deviances in the current and previous C:N:P compilation: SST and nitrate were the most critical drivers in the high latitudes. At the same time, the interaction between nutrient availability and nutrient limitation were the primary drivers in the low latitudes.

**Future projections of ecosystem C:N:P**. We first derived the global GAM formulation of C:P and N:P, covering the entire parameter space of SST, surface nitrate, nutricline, and nutrient limitation. We supplemented POM-metagenome paired samples with 46 POM-only samples collected in high latitudes poleward of 51.5 °S. In doing so, we assumed that these 40 samples were collected from Fe-limited regions based on a comparison with CESM model output (Supplementary Fig. 1a) and prior biogeochemical knowledge[25].

To evaluate the effects of future climatic change on surface community C:P and N:P, we used as input to our GAM derived above the values of SST, surface nitrate concentration, nutricline depth, and nutrient limitation output from CESM2-LENS, which consists of 100 ensemble model simulations which take into the account of the ocean and atmospheric interannual variabilities. The ensemble simulation includes four independent Atlantic Meridional Overturning Circulation states and 20 microstates for each scenario[32]. At the time of writing this paper, 90

out of 100 model outputs were publicly available, and we extracted environmental variables for each grid cell for each of the 90 model run and computed ensemble means for the historic period (averaged values for the years 2010–2014) and the end of the 21st century (averaged values for years 2095–2099), the latter considering Shared Socioeconomic Pathway SSP3-7.0 scenario. SSP3-7.0 scenario is the second most pessimistic, high-greenhouse-gas emission trajectory[33], where $CO_2$ doubles compared to preindustrial by 2100 and radiative forcing level reaches 7.0 W/m². To obtain ensemble mean SST and surface nitrate concentrations for each grid point, we first computed mean values in the top 30 m for each grid point of every model realization and computed the ensemble mean. In each model realization, nutricline was determined first by interpolating the vertical depth profile of nitrate to 1 m in the top 500 m of the water column, then the shallowest depth at which nitrate concentration exceeds 1 μmol kg⁻¹ was determined. After the initial inspection, we found that the nutricline depth obtained from CESM2-LENS systematically underestimated GLODAP. Thus, we multiplied nutricline depth by the scaling factor of 1.54 for every grid point for historical and future projections. The coefficient of determination between GLODAP and CESM2 historic nutricline depth was 0.8.

The limiting nutrient for each grid point is the element with the lowest ratio between the ambient nutrient concentration and the Michaelis-Menten half-saturation constant of the respective element for the small phytoplankton functional type. We defined P/N co-limitation when the ratios between the ambient nutrient concentration and the Michaelis-Menten half-saturation constant for P and N are within 5% and are not Fe-limited. As the nutrient limitation information is a discrete, categorical variable, we computed the ensemble mode across 90 model runs as the representative nutrient limitation for each grid point. The nutrient limitation map from CESM2-LENS for the historic period generally agreed well with the metagenome-based observation[31] (Supplementary Fig. 1a).

To ensure the reliability of our projections, we generated 1000 historic and future C:P and N:P models from the posterior distribution and randomly selected 2000 models with replacements to account for the uncertainties in the parameters of the GAMs. Here, we report averaged predictions from these 2000 models, and we define model confidence by calculating how many of the 2000 pairs of model projections predict the same sign of change in ΔC:P and ΔN:P from the 2010s to 2090s. For example, if all 2000 randomly selected pairs predict an increase (decrease) in C:P, the model confidence is 100%+ (100%-). The null case (i.e., 50% model confidence) is when half of the model pairs predicted an increase, and the other half predicted a decrease. Note that the model uncertainty only considers the uncertainties in the parameters of GAMs, not the variance associated with the ensembled environmental variables from the CESM2-LENS output.

We compared future projections of C:P from the data-derived statistical model with three previously published prognostic ocean biogeochemical outputs under future climate scenarios (Supplementary Fig. 6). These were (1) Minnesota Earth System Model for Ocean biogeochemistry version 3 (MESMO3) under SSP2 scenario[34], (2) Minnesota Earth System Model for Ocean biogeochemistry version 2 (MESMO2) under RCP8.5 scenario[35], (3) Pelagic Interactions Scheme for Carbon and Ecosystem Studies Quota (PISCES-QUOTA) ocean biogeochemistry model under RCP8.5 scenario[14].

**Reporting summary**. Further information on research design is available in the Nature Research Reporting Summary linked to this article.

## Data availability

POM, hydrography, and metagenomes from Bio-GO-SHIP cruises used in this study are publicly available[28,73]. Nutrient stress data of phytoplankton can be accessed from the original publication cited in the main text[31]. GLODAP version2.2016b data is publicly available (https://doi.org/10.5194/essd-8-297-2016). The model output from the CEMS2 Large Ensemble Simulation is available here (https://doi.org/10.26024/kgmp-c556).

## Code availability

All codes (data manipulation, analyses, figures, and tables) can be downloaded from the GitHub repository https://github.com/tanio003/CNPGlobal_paper_repo/tree/CommsEarthEnv. When using the data or code from this project, please cite https://doi.org/10.5281/zenodo.7076407.

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

## Acknowledgements

We thank the many contributing GO-SHIP researchers for the oceanographic data; J.K. Moore and N.A. Wiseman for sharing CESM simulation results; G. Hagstrom and F. Primeau for providing technical advice; and K. Matsumoto and E. Galbraith for valuable comments and suggestions. We gratefully acknowledge financial support by the Simons Foundation (Postdoctoral Fellowship in Marine Microbial Ecology Award 724483 to T.T.), NOAA (101813-Z7554214 to A.C.M. and NOAA Cooperative Institutes, Award #NA19NES4320002, at the Cooperative Institute for Satellite Earth System Studies), NASA (FINESST to C.A.G. and 80NSSC21K1654 to A.C.M.), and the National Science Foundation (OCE-1046297, 1559002, 1848576, and 1948842 to A.C.M.). The PML AMT is funded by the UK Natural Environment Research Council through its National Capability Long-term Single Centre Science Program, Climate Linked Atlantic Sector Science (grant number NE/R015953/1). This study contributes to the international IMBeR project and is contribution number 376 of the AMT programme.

## Author contributions

T.T. compiled metadata, conducted data analysis, and wrote the manuscript with substantial input from all co-authors. C.A.G. coordinated sample collection, processed samples, and compiled metadata. A.A.L. coordinated sample collection, processed samples, and compiled metadata. N.S.G. coordinated sample collection. A.J.F. processed samples. A.C.M. designed and supervised the study, secured funding, and coordinated the Bio-GO-SHIP program.

## Competing interests

The authors declare no competing interests.
