## [Peer Review File · Communications Earth & Environment]

1st Sep 22

Dear Dr Martiny,

Please accept my apologies for the delay in obtaining reports on your manuscript titled "Global patterns and drivers of C:N:P in marine ecosystems". Your manuscript has now been seen by one of the original reviewers (#2) and two new replacement reviewers (#4 and #5), and I include their comments at the end of this message. They find your work of interest, but some important points are raised. We are interested in the possibility of publishing your study in Communications Earth & Environment, but would like to consider your responses to these concerns and assess a revised manuscript before we make a final decision on publication.

We therefore invite you to revise and resubmit your manuscript, along with a point-by-point response that takes into account the points raised. We require that you clearly communicate the caveats and limitations of your analysis and tone down claims where appropriate. We recommend that you evaluate your future predictions derived from contemporary statistical relationships against projections from a global ocean biogeochemical model(s) or, alternatively, you must clearly communicate the caveats and limitations of your approach. We also ask that you are mindful of the language used when interpreting statistical relationships; for example, predictors versus regulators etc. Please highlight all changes in the manuscript text file.

Please use the following link to submit your revised manuscript, point-by-point response to the referees' comments (which should be in a separate document to any cover letter) and the completed checklist:

[link redacted]

We hope to receive your revised paper within six weeks; please let us know if you aren't able to submit it within this time so that we can discuss how best to proceed. If we don't hear from you, and the revision process takes significantly longer, we may close your file. In this event, we will still be happy to reconsider your paper at a later date, as long as nothing similar has been accepted for publication at Communications Earth & Environment or published elsewhere in the meantime.

We understand that due to the current global situation, the time required for revision may be longer than usual. We would appreciate it if you could keep us informed about an estimated timescale for resubmission, to facilitate our planning. Of course, if you are unable to estimate, we are happy to accommodate necessary extensions nevertheless.

Please do not hesitate to contact me if you have any questions or would like to discuss these revisions further. We look forward to seeing the revised manuscript and thank you for the opportunity to review your work.

Best regards,

Clare

Clare Davis, PhD
Senior Editor
Communications Earth & Environment

www.nature.com/commsenv/
@CommsEarth

EDITORIAL POLICIES AND FORMATTING

Editorial Policy: [Policy requirements](https://www.nature.com/documents/nr-editorial-policy-checklist.zip)

Furthermore, please align your manuscript with our format requirements, which are summarized on the following checklist:

[Communications Earth & Environment formatting checklist](https://www.nature.com/documents/commsj-phys-style-formatting-checklist-article.pdf)

and also in our style and formatting guide [Communications Earth & Environment formatting guide](https://www.nature.com/documents/commsj-phys-style-formatting-guide-accept.pdf).

***** DATA:** Communications Earth & Environment endorses the principles of the Enabling FAIR data project (<http://www.copdess.org/enabling-fair-data-project/>). We ask authors to make the data that support their conclusions available in permanent, publically accessible data repositories. (Please contact the editor if you are unable to make your data available).

All Communications Earth & Environment manuscripts must include a section titled "Data Availability" at the end of the Methods section or main text (if no Methods). More information on this policy, is available at <http://www.nature.com/authors/policies/data/data-availability-statements-data-citations.pdf>.

If a community resource is unavailable, data can be submitted to generalist repositories such as [figshare](https://figshare.com/) or [Dryad Digital Repository](http://datadryad.org/). Please provide a unique identifier for the data (for example a DOI or a permanent URL) in the data availability statement, if possible. If the repository does not provide identifiers, we encourage authors to supply the search terms that will return the data. For data that have been obtained from publically available sources, please provide a URL and the specific data product name in the data availability statement. Data with a DOI should be further cited in the methods reference section.

REVIEWER COMMENTS:

Reviewer #2 (Remarks to the Author):

The authors have sufficiently responded to my comments from the previous review for their submission to Nature Geoscience. I am especially pleased to see the previous supplemental figures now incorporated into Fig. 4. I do think those results are particularly compelling and commend the authors for focusing more on this aspect of the narrative.

I have only one small suggestion: It would be helpful to qualify what is meant by "future climate projections" in the abstract if room allows, since there are so many ways in which one might carry out these projections. Perhaps something along the lines of "Future climate projections using a data-derived statistical model...".

Reviewer #4 (Remarks to the Author):

The manuscript of Tanioka et al. presents a new analysis of the elemental stoichiometry of marine particulate organic matter (POM), relating this in a statistical sense to a number of potential drivers of the observed stoichiometric variability. The problem being tackled is important and there is considerable value within both the new data presented and new analyses provided within the manuscript. Indeed, further expansion of high quality consistent data set on marine POM stoichiometry in itself would be of value, while the additional statistical analysis represents a reasonable attempt to further test well explored hypotheses.

I note that the manuscript is a re-submission having been previously submitted to Nature Geoscience. As far as I can tell, the authors appear to have done a reasonable job of addressing the majority of the comments of the original reviewers, although I note that there is at least one significant comment which I thought could have been more fully addressed (see below).

Overall I have a number of comments that I would like to see the authors address in a further submission.

General comments:

The analysis performed indicates the environmental / ecosystem variables which are predictors of the stoichiometric ratios. As the authors indicate in the response to the original reviewers, these predictive/correlative relationships may well be causative, particularly as they are consistent with a range of existing hypotheses, however I would encourage the authors to more clearly outline to the reader the difference between statistical relationships and the associated interpretation of these. Again, I note similar points were made in the previous set of reviews. I further note that some of the relationships may still not be causative, at least in a direct sense, but may rather reflect underlying indirect relationships between variables. For example, SST will ultimately be determined by both heat exchange at the surface of the ocean and interior oceanic circulation, both of which could change the dynamics of marine ecosystems indirectly.

Partly associated with above, I agree with the comments of previous reviewer 3 around the uncertainties / caveats involved in projections of future changes on the basis of the statistical analysis. There are significant caveats associated with this as there is no reason to assume that the statistical relationships revealed in the current analysis, which are by definition a representation of the contemporary state of the system, will necessarily persist under an altered state. For example, it is entirely possible that the statistical relationships between temperature and nutrient stress may vary into the future. Warming temperatures will be a direct consequence of future increases in radiative forcing. Although this warming may also influence circulation and hence nutrient supply and hence ultimately limitation, there is no reason to assume that the statistical relationships between warming and nutrient limitation and supply will remain the same. Although some caveats are already discussed from Line 203 onwards, I would like to see the authors at least acknowledge and highlight to the reader this potential fundamental issue with the types of projections performed which try and make predictions outside of the current (statistical state of) the

system and also (presumably) outside of the observed state of the current system (e.g. for SST >30°C).

Further specific comments:

Line 17: Rephrase (suggest remove 'the detailed')

Line 21: suggest change 'are responsible for' to 'are predictive of'

Line 22: suggest 'allowed us to diagnose community nutrient stress types'

Line 24: see substantial point above, the future predictions were not so convincing, but if they are to remain associated caveats should be more clearly stated.

Line 41: I am not sure the alternative hypotheses 'compete' as most are not mutually exclusive. Suggest 'There are several alternate, although not necessarily mutually exclusive, hypotheses...'

Line 81 (and elsewhere): be consistent between C:N:P and CNP (suggest former throughout)

Line 100: suggest 'here defined as the depth at which nitrate...'. See also Line 466.

Line 104: the data used to infer species composition has not been stated prior to this point.

Line 104: rephrase 'Nitrate and phosphate concentrations...'

Line 110: 'in the high latitudes'

Line 125: suggest 'when nutricline depth approached 0 m and thus where nitrate remained abundant at the surface.'

Line 206: I think this should be 'low temperature and high macronutrient stress', as high latitudes (particularly in the south) will be characterised by low temperatures and high iron stress.

Line 208: similar to above 'Arctic Ocean regions with low temperature and depressed macronutrients...'

Line 541: First introduction of different models in text is here. All models should be introduced at same time.

Fig. 3 caption: suggest 'predictors of ecosystem C:N:P'

Reviewer #5 (Remarks to the Author):

For the review, please see the attached word document.

Tanioka et al. – Global patterns and drivers of C:N:P in marine ecosystems.

I have been asked by the editor to stand in for the previous reviewer 3, as they could not offer a second report. I have therefore been brought on to assess if the authors have acknowledged and addressed the concerns of the previous reviewer 3.

I will address each concern below, with my opinion in red.

Concerns of previous Reviewer 3

1. That the paper is better suited to a longer format, specialist journal given the caveats and uncertainties in the approach.
Agreed, and the authors also agree.
2. Lack of data and therefore an undermining of the confidence with which the authors present their interpretation.
I feel that the authors remain too strong in their interpretations. For instance, on line 168, the authors state “In low latitude ecosystems, we observe a strong regulation from the interaction between nutricline depth and elemental nutrient stress type.” This is much too strong for an explained deviance in the GAM analysis of < 0.4, even 0.13 for C:N ratios. That is weak evidence for regulation, if anything.
3. A “mix and match” approach of data sources introduces substantial uncertainty. The C:N:P, hydrography, nutrients and nutrient-stress genes come from *in situ* sampling, mixed layer PAR comes from a climatology, phytoplankton community composition comes from the NASA Ocean Biogeochemical Model, and total dissolved iron (FeT) comes from the Community Earth System Model. While the *in situ* samples are great, the supplementation with model data is not, and these models will likely not be doing a good job with FeT concentrations.
This is less of a concern to me, and I feel that the authors have addressed this.
4. No consideration of how climatologies or models stack up against observations.
Again, acknowledged and addressed.
5. Omission of dissolved forms of nutrients for influencing C:N:P ratios.
This is a clear omission and is not addressed in the discussion. I think it is easily remedied though.
6. Calculation of nitracline is unusual.
*It is odd to me that so many places in the high latitudes there are now non-zero values for this metric. If NO_3 is > 0 at the surface, then their values should be zero. But there are clearly many non-zero values in Figure 3 for Fe-limited regions.
I would also take the opportunity to say that it is unusual that the depth of the nitracline is such an important predictor of C:N:P in the high latitudes because this depth should almost universally be near zero based on their method of calculating it. Furthermore, it isn't clear to me, mechanistically, why nutricline depth should control C:N:P in the high latitudes. The authors (1) do not acknowledge this in the discussion and (2) do not address why nitracline*

depth is a more powerful predictor of C:N:P ratios than SST in their GAMS! Instead they say that SST is the most important predictor, and this is not true to their results.

7. Use of only *Prochlorococcus* genomes for estimates of nutrient stress.
This should be acknowledged in the discussion.
8. Unconvinced of GAM statistical model for predicting C:N:P ratios using key environmental variables.
This is a big concern. The way the authors present the results of the GAM is not completely honest. It is all presented in “normalised explained deviance”, such that “In these warm regions, we observe that 77%-87% of the explained variance for C:N:P was attributed to the nutricline depth plus element-specific nutrient stress”. This is fine in the high latitudes where $R^2 > 0.5$, but the explained deviance is very low in the low latitudes with $R^2 < 0.4$ (even as low as 0.13), so that sentence is very misleading. The authors should be up front with the limitations of the GAM and mention the R^2 values directly in the text.
9. “Extension of the statistical approach to the future ... seems at best premature and potentially also misleading”. Reviewer encourages the authors to compare their statistical predictions with those of a prognostic ocean biogeochemical model.
And, if you read on from my previous paragraph, the limitations of the GAM directly undermine the predictions made in the future scenarios. I have to agree with the reviewer. At the very least, the authors should try to compare their statistical predictions with the predictions from a global ocean biogeochemical model with variable stoichiometry, of which there are a few now in operation.

Summary

Tanioka et al. have acknowledged and addressed some of the concerns of Reviewer #3, but not all of them. I would also say that their language around the strength of the results is misleading. I would encourage revisions.

Response to editor's and reviewer's comments. Reviewer comments are in black font, and our responses are in blue. Line numbers correspond to those used in the **cleaned** manuscript file.

Response to Reviewer #2

The authors have sufficiently responded to my comments from the previous review for their submission to Nature Geoscience. I am especially pleased to see the previous supplemental figures now incorporated into Fig. 4. I do think those results are particularly compelling and commend the authors for focusing more on this aspect of the narrative.

I have only one small suggestion: It would be helpful to qualify what is meant by "future climate projections" in the abstract if room allows, since there are so many ways in which one might carry out these projections. Perhaps something along the lines of "Future climate projections using a data-derived statistical model...".

We appreciate reviewer #2 for spending their time and effort to review our manuscript in the resubmission. In accordance with their suggestion, we have rephrased a sentence in the abstract now read "Future climate projections using a data-derived statistical model suggest that C:P and N:P ratios will increase at high latitudes, but changes are uncertain at low latitudes due to a lack of observations at extreme surface ocean temperature and poorly constrained shifts in N vs. P stress."

Response to Reviewer #4

The manuscript of Tanioka et al. presents a new analysis of the elemental stoichiometry of marine particulate organic matter (POM), relating this in a statistical sense to a number of potential drivers of the observed stoichiometric variability. The problem being tackled is important and there is considerable value within both the new data presented and new analyses provided within the manuscript. Indeed, further expansion of high quality consistent data set on marine POM stoichiometry in itself would be of value, while the additional statistical analysis represents a reasonable attempt to further test well explored hypotheses.

I note that the manuscript is a resubmission having been previously submitted to Nature Geoscience. As far as I can tell, the authors appear to have done a reasonable job of addressing the majority of the comments of the original reviewers, although I note that there is at least one significant comment which I thought could have been more fully addressed (see below).

Overall I have a number of comments that I would like to see the authors address in a further submission.

We thank reviewer #4 for assessing our manuscript and exposing some key issues that we now addressed in this revision.

General Comments:

The analysis performed indicates the environmental / ecosystem variables which are predictors of the stoichiometric ratios. As the authors indicate in the response to the original reviewers, these predictive/correlative relationships may well be causative, particularly as they are consistent with a range of existing hypotheses, however I would encourage the authors to more clearly outline to the reader the difference between statistical relationships and the associated interpretation of these. Again, I note similar points were made in the previous set of reviews. I further note that some of the relationships may still not be causative, at least in a direct sense, but may rather reflect underlying indirect relationships between variables. For example, SST will ultimately be determined by both heat exchange at the surface of the ocean and interior oceanic circulation, both of which could change the dynamics of marine ecosystems indirectly.

We agree with the notion that the distinction between causal and correlation relationships is hard to disentangle in natural settings. We now attempt to clarify and distinguish statistical relationships from the causal relationship by using the term “predictors” instead of “regulators” or “drivers.” We also now state upfront in the introduction the potential advantages and limitations of data-driven statistical approaches: “The data-driven statistical approach, which first establishes relationships amongst C:N:P and environmental factors along contemporary ocean environmental gradients and then applies the same statistical relationship to the future environmental condition, is an alternative to Earth system models for predicting future changes to C:N:P. Although data-driven statistical approaches lack a mechanistic basis, they can integrate poorly understood biological mechanisms. For example, this approach implicitly

‘embraces’ the plankton diversity, interactions between different environmental factors, and poorly understood biotic effects of higher trophic levels³⁶.” (lines 80-87).

Partly associated with above, I agree with the comments of previous reviewer 3 around the uncertainties / caveats involved in projections of future changes on the basis of the statistical analysis. There are significant caveats associated with this as there is no reason to assume that the statistical relationships revealed in the current analysis, which are by definition a representation of the contemporary state of the system, will necessarily persist under an altered state. For example, it is entirely possible that the statistical relationships between temperature and nutrient stress may vary into the future. Warming temperatures will be a direct consequence of future increases in radiative forcing. Although this warming may also influence circulation and hence nutrient supply and hence ultimately limitation, there is no reason to assume that the statistical relationships between warming and nutrient limitation and supply will remain the same. Although some caveats are already discussed from Line 203 onwards, I would like to see the authors at least acknowledge and highlight to the reader this potential fundamental issue with the types of projections performed which try and make predictions outside of the current (statistical state of) the system and also (presumably) outside of the observed state of the current system (e.g. for SST >30 degC).

We thank the reviewer for these critical insights and now acknowledge and highlight the fundamental limitation with our approach in the last paragraph (lines 234-238): “There are several important caveats to our new observation and the data-driven statistical approach for projecting C:N:P. First, data-driven statistical models assume that plankton physiology and community will share the same relationship to environmental conditions present and future. These projections incur significant uncertainties when extrapolating the statistical models outside the currently observed/observable state of the system.”

Further specific comments:

Line 17: Rephrase (suggest remove ‘the detailed’)

DONE as suggested

Line 21: suggest change ‘are responsible for’ to ‘are predictive of’

DONE as suggested

Line 22: suggest ‘allowed us to diagnose community nutrient stress types’

DONE as suggested

Line 24: see substantial point above, the future predictions were not so convincing, but if they are to remain associated caveats should be more clearly stated.

DONE. Please refer to our response to the general comments.

Line 41: I am not sure the alternative hypotheses ‘compete’ as most are not mutually exclusive. Suggest ‘There are several alternate, although not necessarily mutually exclusive, hypotheses...’

DONE as suggested.

Line 81 (and elsewhere): be consistent between C:N:P and CNP (suggest former throughout)
We “CNP” changed to “C:N:P” in all instances where it means ratio. CNP is retained when specifically referring to “Carbon-Nitrogen-Phosphorus cycle” in the abstract.

Line 100: suggest ‘here defined as the depth at which nitrate...’. See also Line 466.
DONE as suggested

Line 104: the data used to infer species composition has not been stated prior to this point.
DONE. We added a sentence “Here, phytoplankton-group relative abundance was obtained from the NASA Ocean Biogeochemical Model (NOBM)^{38,39} at the closest grid point to the spatial position of each POM sampling point.” (lines 124-126)

Line 104: rephrase ‘Nitrate and phosphate concentrations...’
DONE as suggested.

Line 110: ‘in the high latitudes’
DONE as suggested.

Line 125: suggest ‘when nutricline depth approached 0 m and thus where nitrate remained abundant at the surface.’
DONE as suggested.

Line 206: I think this should be ‘low temperature and high macronutrient stress’, as high latitudes (particularly in the south) will be characterised by low temperatures and high iron stress.
We agree and changed as suggested.

Line 208: similar to above ‘Arctic Ocean regions with low temperature and depressed macronutrients...’
DONE as suggested.

Line 541: First introduction of different models in text is here. All models should be introduced at same time.

We now introduce all the six possible hierarchical GAM candidates at the same time. “to determine the best model out of the six possible hierarchical GAM model formulations: (1) Model G (A global smoother for all observations), (2) Model GS (Single common smoother plus group-level smoothers that have the same wigginess), (3) Model GI (Single common smoother plus group-level smoothers that have the different wigginess), (4) Model S (Group-specific smoothers without a global smoother, but all smoothers have the same wigginess), (5) Model I (Group-specific smoothers with different wigginess), and (6) Model C (Control, no dependence on nutrient limitation types) (Supplementary Information).” (lines 595-602)

Fig. 3 caption: suggest 'predictors of ecosystem C:N:P'
DONE as suggested

Response to Reviewer #5

I have been asked by the editor to stand in for the previous reviewer 3, as they could not offer a second report. I have therefore been brought on to assess if the authors have acknowledged and addressed the concerns of the previous reviewer 3. [I will address each concern below, with my opinion in red.]

We greatly thank reviewer #5 for accepting to newly review our manuscript and providing feedback.

Concerns of previous Reviewer 3

1. That the paper is better suited to a longer format, specialist journal given the caveats and uncertainties in the approach. Agreed, and the authors also agree.

As resolved in the previous revision

2. Lack of data and therefore an undermining of the confidence with which the authors present their interpretation. I feel that the authors remain too strong in their interpretations. For instance, on line 168, the authors state “In low latitude ecosystems, we observe a strong regulation from the interaction between nutricline depth and elemental nutrient stress type.”. This is much too strong for an explained deviance in the GAM analysis of < 0.4 , even 0.13 for C:N ratios. That is weak evidence for regulation, if anything.

We now toned-down interpretation of our results in the discussion section regarding C:N:P in low latitude ecosystems. These are:

1. “In low latitude ecosystems, our global data suggest C:N:P is regulated in part by an interaction between the overall nutrient supply and the elemental nutrient stress type.” (lines 196-197).
2. “Thus, the observed interactive relationships between C:N:P, nutricline depth, and N vs. P stress seem to align well with these theoretical and laboratory culture predictions.” (lines 203-205)
3. “In summary, nutrient supply rate and ratios are potentially the best predictors of large C:N:P variability in low latitude marine ecosystems, while temperature and macronutrient availability seem to shape the overall latitudinal gradient.” (lines 220-222).

3. A “mix and match” approach of data sources introduces substantial uncertainty. The C:N:P, hydrography, nutrients and nutrient-stress genes come from in situ sampling, mixed layer PAR comes from a climatology, phytoplankton community composition comes from the NASA Ocean Biogeochemical Model, and total dissolved iron (FeT) comes from the Community Earth System Model. While the in situ samples are great, the supplementation with model data is not, and these models will likely not be doing a good job with FeT concentrations. This is less of a concern to me, and I feel that the authors have addressed this.

As resolved in the previous revision

4. No consideration of how climatologies or models stack up against observations. Again, acknowledged and addressed.

As resolved in the previous revision

5. Omission of dissolved forms of nutrients for influencing C:N:P ratios. This is a clear omission and is not addressed in the discussion. I think it is easily remedied though.

We now include the discussion of DOP in the discussion section: “Second, we did not consider the roles of dissolved organic matter (DOM). Plankton’s ability to access DOM, particularly at high temperatures, may be an important driver for shifting the balance between C, N, and P in areas such as North Atlantic and western South and North Pacific⁵⁰. However, DOM is chemically diverse⁵¹ and we were unable to incorporate DOM as a predictor here.” (lines 238-242)

6. Calculation of nitracline is unusual. It is odd to me that so many places in the high latitudes there are now non-zero values for this metric. If NO₃ is > 0 at the surface, then their values should be zero. But there are clearly many non-zero values in Figure 3 for Fe-limited regions. I would also take the opportunity to say that it is unusual that the depth of the nitracline is such an important predictor of C:N:P in the high latitudes because this depth should almost universally be near zero based on their method of calculating it.

We use nutricline from in situ bottle data instead of GLODAP data resulting in more nonzero in the high latitudes. This is an artifact of how we defined nutricline depth; we were assigning nutricline depth as the shallowest bottle depth (which ranges between 0-20 m) and not at 0 m, in which regions where NO₃ > 0 at the shallowest bottle depth. Following suggestion, we set nutricline as 0 m when the bottle NO₃ concentration at the shallowest depth is greater than 1 μmol/kg. “Nutricline depth, here defined as the depth at which nitrate equals 1 μmol kg⁻¹, was determined by vertically and horizontally interpolating nitrate concentration. We set nutricline as 0 m when the bottle nitrate concentration at the shallowest depth was greater than 1 μmol kg⁻¹.” (lines 522-525).

We greatly thank the reviewer for spotting this.

Furthermore, it isn’t clear to me, mechanistically, why nutricline depth should control C:N:P in the high latitudes. The authors (1) do not acknowledge this in the discussion and (2) do not address why nitracline depth is a more powerful predictor of C:N:P ratios than SST in their GAMS! Instead they say that SST is the most important predictor, and this is not true to their results.

After correcting for the nutricline depth calculation, we have reconducted analyses and found that nutricline indeed does not control C:N:P in high latitudes (Fig. 3b TOP).

7. Use of only *Prochlorococcus* genomes for estimates of nutrient stress. This should be acknowledged in the discussion.

We now mention this in discussion section: “Thirdly, we solely used *Prochlorococcus* genomes to diagnose nutrient stress for the plankton community. As *Prochlorococcus* make up a large percentage of community biomass in the tropics and subtropics⁵², their physiological status is likely important for the total phytoplankton community. However, in regions with lower

Prochlorococcus abundance, other lineages are likely important for the ecosystem state and may deviate from *Prochlorococcus*.” (lines 242-247)

8. Unconvinced of GAM statistical model for predicting C:N:P ratios using key environmental variables. This is a big concern. The way the authors present the results of the GAM is not completely honest. It is all presented in “normalised explained deviance”, such that “In these warm regions, we observe that 77%-87% of the explained variance for C:N:P was attributed to the nutricline depth plus element-specific nutrient stress”. This is fine in the high latitudes where $R^2 > 0.5$, but the explained deviance is very low in the low latitudes with $R^2 < 0.4$ (even as low as 0.13), so that sentence is very misleading. The authors should be up front with the limitations of the GAM and mention the R^2 values directly in the text.

Following the suggestion, we now provide R^2 directly in the main text: “In these warm regions, we observed that 77 - 87% of the explained variance for C:N:P was attributed to the nutricline depth plus element-specific nutrient stress (Fig. 3b). However, total deviance explained by GAM was noticeably lower in the low latitude ecosystems ($R^2 = 0.39, 0.37,$ and 0.14 for C:P, N:P, and C:N) than in the high latitude ecosystems.” (lines 139-143)

9. “Extension of the statistical approach to the future ... seems at best premature and potentially also misleading”. Reviewer encourages the authors to compare their statistical predictions with those of a prognostic ocean biogeochemical model. And, if you read on from my previous paragraph, the limitations of the GAM directly undermine the predictions made in the future scenarios. I have to agree with the reviewer. At the very least, the authors should try to compare their statistical predictions with the predictions from a global ocean biogeochemical model with variable stoichiometry, of which there are a few now in operation. We now include a figure comparing our projection with ocean BGC models in the Supplementary Fig. 6 (attached below) and discussed similarities and differences in the results section (lines 156-178).

Supplementary Fig. 6: Projections of change in C:P under future conditions for different models. (a) % change in C:P projected with data-driven GAM applied to output from CESM2 Large Ensemble simulation under SSP3-7.0 scenario from the 2010s to 2090s, (b) % change in model C:P export ratio of POM at 100 m under SSP2 scenario from 2000 to 2100 with MESMO3³⁴, (c) % change in model suspended C:P ratio of POM at 100 m under RCP8.5 scenario from 2010s to 2090s with PISCES-QUOTA¹⁵, (d) % change in model C:P export at 100 m under RCP8.5 scenario from 2000 to 2090s with MESMO2³⁵.

Summary

Tanioka et al. have acknowledged and addressed some of the concerns of Reviewer #3, but not all of them. I would also say that their language around the strength of the results is misleading. I would encourage revisions.

We thank the reviewer once again, and we now believe that all the issues have been resolved.

5th Oct 22

Dear Dr Martiny,

Your manuscript titled "Global patterns and drivers of C:N:P in marine ecosystems" has now been seen by our reviewers, whose comments appear below. In light of their advice I am delighted to say that we are happy, in principle, to publish a suitably revised version in Communications Earth & Environment under the open access CC BY license (Creative Commons Attribution v4.0 International License).

We therefore invite you to revise your paper one last time to comply with our format requirements and to maximise the accessibility and therefore the impact of your work.

Please note that it may still be possible for your paper to be published before the end of 2022, but in order to do this we will need you to address these points as quickly as possible so that we can move forward with your paper.

EDITORIAL REQUESTS:

SUBMISSION INFORMATION:

OPEN ACCESS:

Communications Earth & Environment is a fully open access journal. Articles are made freely accessible on publication under a [CC BY license](http://creativecommons.org/licenses/by/4.0) (Creative Commons Attribution 4.0 International License). This license allows maximum dissemination and re-use of open access materials and is preferred by many research funding bodies.

For further information about article processing charges, open access funding, and advice and support from Nature Research, please visit <https://www.nature.com/commsenv/article-processing-charges>

At acceptance, you will be provided with instructions for completing this CC BY license on behalf of all authors. This grants us the necessary permissions to publish your paper. Additionally, you will be asked to declare that all required third party permissions have been obtained, and to provide billing

information in order to pay the article-processing charge (APC).

[link redacted]

Best regards,

Clare

Clare Davis, PhD
Senior Editor
Communications Earth & Environment

www.nature.com/commsenv/
@CommsEarth

REVIEWERS' COMMENTS:

Reviewer #4 (Remarks to the Author):

The authors have done a reasonable job of responding to my original review. In particular I was pleased to see a much more thorough description of the caveats associated with extrapolation of the statistical results to derive future projections. I also think the manuscript has been considerably improved by consideration of the predictor variables as such (rather than drivers). I am now supportive of publication.

One final minor comment, suggest ammend Line 422: '... to environmental conditions in the present and future ocean'

Reviewer #5 (Remarks to the Author):

I am satisfied that the authors have addressed and acknowledged the concerns of both myself and the other reviews. It is also commendable that they included the biogeochemical model output for comparison with their statistical model.

Response to reviewer's comments. Reviewer comments are in black font, and our responses are in blue. Line numbers correspond to those used in the **cleaned** manuscript file.

Response to Reviewer #4

The authors have done a reasonable job of responding to my original review. In particular I was pleased to see a much more thorough description of the caveats associated with extrapolation of the statistical results to derive future projections. I also think the manuscript has been considerably improved by consideration of the predictor variables as such (rather than drivers). I am now supportive of publication.

One final minor comment, suggest ammend Line 422: '... to environmental conditions in the present and future ocean'

We appreciate reviewer #4 once again for spending their time and effort to review our manuscript in the resubmission.

We have amended the line mentioned above in the manuscript, and it now reads: "First, data-driven statistical models assume that plankton physiology and community will share the same relationship to environmental conditions in the present and future ocean." (lines 236-237).

Response to Reviewer #5

I am satisfied that the authors have addressed and acknowledged the concerns of both myself and the other reviews. It is also commendable that they included the biogeochemical model output for comparison with their statistical model.

We thank reviewer #5 for reviewing our manuscript, and we are glad we were able to address the issues.